# Association of Protein Intake with Handgrip Strength and Its Relation to Strength Exercise in Korean Adults Aged over 60 Years in the KNHANES (2014-18)

**DOI:** 10.3390/nu15041014

**Published:** 2023-02-17

**Authors:** Eun Young Choi

**Affiliations:** Department of Family Medicine, School of Medicine, Dankook University, 119 Dandae-ro, Dongnam-gu, Cheonan City 31116, Chungnam, Republic of Korea; choiey@dku.edu or choiey0410@gmail.com; Tel.: +82-41-550-3998; Fax: +82-0504-015-2053

**Keywords:** handgrip strength, dietary protein intake, strength exercise, Korean

## Abstract

Weak handgrip strength (HGS) is associated with many negative health outcomes in older adults. There is evidence that with strength exercise, high protein intake leads to increased HGS. The goal of this study was to examine the relationship between weak HGS in older adults and dietary protein and it’s relation to resistance exercise. Data on 8497 Korean adults aged over 60 years from the Korea National Health and Nutrition Examination Survey (2014-18) were analyzed. Dietary protein intake measured by 24-h recall were categorized as three levels: low (<0.8 g/kg body weight (BW)), adequate (≥0.8 g/kg BW and <1.2 g/kg BW), and high (≥1.2 g/kg BW). Complex sample multiple logistic regression analyses were carried out. The prevalence of weak HGS was 18.3 (0.7)% in men and 28.8 (0.9)% in women. In a multiple logistic regression analysis, low protein intake was significantly associated with higher risk of weak HGS in men, as compared with adequate protein intake, but this relationship was not statistically significant in women. The risk of weak HGS was significantly reduced for both men and women who engaged in strength exercise and increased their dietary protein intake. In this study, based on a representative sample of Koreans aged over 60, men with low protein intake had a higher risk of weak HGS than did men with adequate protein intake. Men and women with a higher protein intake who also engaged in strength exercises had a lower risk of weak HGS. Increasing protein intake and engaging in strength exercises may be an effective way to preserve muscle strength in older men and women.

## 1. Introduction

Handgrip strength (HGS) is an inexpensive and practical tool for evaluating muscle strength, which reflects not only the upper-body strength but also the strength of the lower and central muscles, when measured while standing [1]. Numerous negative health outcomes, such as chronic morbidities, functional impairments, and all-cause mortality, are known to be linked to weak HGS [2]. So, early detection and management of weak HGS is important for prevention of adverse health outcomes.

HGS would greatly improve with training that focuses primarily on upper extremity exercises that involve handgrip movements [3]. However, the results of the meta-analysis regarding the impact of exercise training on HGS in older adults showed that enhancements in HGS were seen, not only from interventions that solely emphasized strength and resistance training, but also from multi-modal approaches [3]. Despite the difficulty in determining the influence of specific training factors, such as mode, duration, or intensity, it is possible that insufficient intervention period, insufficient frequency of training sessions, or insufficient duration of individual training sessions could contribute to the absence of improvement in HGS through exercise intervention [3,4]. In addition to exercise, HGS is known to be related to many factors including age, physical inactivity, comorbid disease, and inadequate nutrition [5]. Among the modifiable risk factors, insufficient dietary protein intake has been identified as critical for maintenance of HGS, because it assures the delivery of necessary amino acids and promotes protein synthesis [6]. Although dietary protein intake of 0.8 g/kg per day is recommended for all adults regardless of age [7], there is mounting evidence suggesting that increased protein consumption may help prevent age-related muscle wasting and preserve muscular function [8,9]. However, a recent meta-analysis showed that protein has no further benefits for altering HGS in older adults without resistance exercise [10].

Previous Korean studies investigating protein intake with weak HGS were inconsistent in their results [11,12,13]; generally, there is also a lack of research evaluating the relationship between dietary protein intake and weak HGS in Korean older adults with consideration of resistance exercise.

Therefore, this study aimed to investigate the association of dietary protein intake with weak HGS and its relation to resistance exercise in representative Koreans aged over 60 years, using data from the KNHANES (2014-18).

## 2. Methods

### 2.1. Study Subjects

The data for this study were collected as part of the Korean National Health and Nutrition Examination Survey (KNHANES), a large, nationwide study of the South Korean general population. The survey, conducted by the Korea Centers for Disease Control and Prevention, includes a diverse, representative sample of non-institutionalized South Koreans. The survey uses a complex sampling design that involves stratification and clustering of the samples to ensure that they are representative of the overall population. The data, collected through a rolling survey process over the 2014-18 period, included a total of 39,199 participants. Of these, 10,896 adults 60 years of age or older were chosen for further analysis. Further exclusions, due to participants’ lacking complete information on HGS (*n* = 1362), body weight and height (*n* = 39), or protein intake (*n* = 998), meant that there remained 8497 participants eligible for inclusion in the study (Figure 1). All of the KNHANES participants agreed to the study’s use of their data by signing written consent forms, and, according to the Bioethics and Safety Act, no additional ethical review was necessary.

### 2.2. Anthropometry and HGS Measurements

The participants were dressed simply and without shoes when their weight and height were measured. Body mass index (BMI) was calculated by dividing the weight in kilograms by the squared height in meters. HGS was evaluated three times in each hand using a TKK 5401 digital grip-strength dynamometer (Takei Scientific Instruments Co., Ltd., Tokyo, Japan). Participants were instructed to stand upright with their arms at their sides and squeeze the grip with as much force as possible for three seconds each time. The dominant hand’s maximal measured grip strength from the three trials was determined as the maximal HGS. HGS measurement was not performed on people who had undergone wrist surgery during the preceding three months or reported wrist pain within the last 7 days. Weak HGS was defined as a maximal measured grip strength of less than 28 kg for men and less than 18 kg for women, according to the standards defined by the Asian Working Group for Sarcopenia [13,14].

### 2.3. Protein Intake Assessment

A trained interviewer conducted a home interview one week after the health interview and examination to assess dietary intake using a single 24-h diet recall method. To calculate the amounts of total energy and other macronutrients, the Korean Foods and Nutrients Database of the Rural Development Administration was utilized [15,16]. Based on their reported protein intake, as well as recommendations for older adults from the PROT-AGE study group and the Nordic Nutrition Recommendations [9], the participants were divided into three categories: low protein intake (<0.8 g/kg body weight), adequate protein intake (≥0.8 and <1.2 g/kg body weight), and high protein intake (≥1.2 g/kg body weight).

### 2.4. Physical Activity Assessment

From 2014, the KNHANES began using the Global Physical Activity Questionnaire (GPAQ) in order to more accurately measure and interpret physical activity levels by area and intensity. The GPAQ was translated into Korean by the Korea Disease Control and Prevention Agency in 2013, and its reliability and validity were verified [17]. The GPAQ was utilized to assess the level of moderate and vigorous physical activity across three distinct behavior categories, which include: physical activity at work, transportation-related physical activity, and leisure time physical activity. Time was calculated according to vigorous intensity and moderate intensity activities (the numbers of days of activity per week × number of times (min) of activity per day. Aerobic physical activity was defined as engaging in 150 min of moderate intensity physical activity or 75 min of vigorous intensity physical activity or an equivalent combination of moderate and vigorous intensity physical activity, based on the GPAQ. Participants deemed to be those who perform strength exercise typically engaged in activities, such as push-ups, sit-ups, dumbbells, weights, and horizontal bar more than twice in the past week.

### 2.5. Assessment of Covariates

Current smokers were defined as those who currently smoke and have smoked at least 100 cigarettes in their lifetime. Heavy alcohol drinkers were those whose average daily alcohol consumption was 30 g or more. Education level was classified simply as university graduate or not. Participants deemed to have chronic disease were suffering from, or had suffered, from one of the following: diabetes, heart attack, chronic kidney disease, liver cirrhosis, stroke, osteoarthritis, thyroid disease, asthma, or chronic obstructive pulmonary disease. Participants who responded “yes” when asked if they were currently constrained in their everyday lives and social activities because of health issues or physical or mental diseases were considered to have a limitation. Occupation was divided into two categories: white-collar workers and blue-collar workers. White-collar workers included managers, professionals, clerks, and sales workers. Blue-collar workers included skilled workers in agriculture, forestry, fishing, craft trades, machine operation, and elementary labor.

### 2.6. Statistical Analysis

The KNHANES sampling weights were considered in the complex sample analyses using SPSS software (version 25.0; IBM, Inc., Armonk, NY, USA) to achieve nationally representative estimates. Excluded subjects were regarded as subgroups in order to avoid skewed results. The data for men and women were analyzed separately due to differences in HGS and dietary protein intake. The participants were divided into three groups based on protein intake (<0.8 g/kg body weight (BW), ≥0.8 g/kg BW and <1.2 g/kg BW, ≥1.2 g/kg BW), as already noted. The complex sample chi-square test and linear regression test were used to determine the differences in categorical and continuous variables, respectively, according to protein intake levels. Complex sample multivariate logistic regression analyses were performed to investigate the association between weak HGS and protein intake level, adjusting for sociodemographic factors (age, education, BMI, occupation), lifestyle factors (smoking, heavy alcohol drinking, aerobic physical activity, strength exercise), and comorbidity (presence of limitation and chronic disease). Considering the high correlation between BMI and total energy intake, total energy intake was not included in the models.

The group with adequate protein intake was used as the reference category for all models with categorical protein variables. In the tables with the categorized protein variables, test results for linear trends also are shown when comparing the groups with low and high protein intakes. Additionally, subgroup analysis was performed for strength exercises.

## 3. Results

Among Korean adults aged over 60 years, 18.3 (1.3)% of men had weak HGS, and 28.8 (0.9)% of women. The average daily protein intake was 69.8 (0.4) g and 1.07 (0.00) g/kg for men and 58.1 (0.3) g and 1.04 (0.00) g/kg for women. Low protein intake was identified in 34.5 (1.0)% of men and 49.2 (0.9)% of women. Regardless of gender, those in the high protein intake group were younger, had higher HGS, and lower BMI. They also performed more aerobic and strength exercises, were more likely to be college graduates, white collar workers, and heavy drinkers, and less likely to have chronic disease or limitation. However, the proportion of those “currently smoking” did not differ by protein intake among either men or women (Table 1).

The association between each of the protein intake groups and weak HGS is presented in Table 2. The prevalence of weak HGS was highest in the low protein intake group: 23.7 (1.3)% in men and 33.6 (1.4)% for women. As the level of protein intake increased, the prevalence of weak HGS decreased (*P_trend_* < 0.001) in both men and women. Compared with adequate protein intake, the low protein intake group had a significantly higher odds ratio (OR) of weak HGS: 1.46 (95% confidence interval [CI]:1.16–1.83) in men after adjusting for all covariates. As the level of protein intake decreased, the OR of weak HGS decreased in men (*P_trend_* = 0.035). In women, the odds of weak HGS for low protein intake (OR 1.18, 95% CI: 0.93–1.50) were not significantly higher than for adequate protein intake after adjusting for all covariates. High protein intake was not associated with lower odds of weak HGS in either gender.

As for the subgroup analysis performed for strength exercises, men with low protein intake who engaged in strength exercises had not significantly higher odds of weak HGS (OR 1.77, 95% CI: 0.88–3.54), as compared with men with adequate protein intake after adjusting for all covariates (Table 3). Women with low protein intake who engaged in strength exercises had significantly higher odds of weak HGS, that is, 2.46 (95% CI: 1.0.3–5.92) as compared with women with adequate protein intake after adjusting for all covariates. As the level of dietary protein intake increased, the OR of weak HGS decreased in both men and women with strength exercises (*P_trend_* < 0.001 for men and *P_trend_* = 0.025 for women, respectively). There was no significant association between dietary protein intake and weak HGS in both men and women who did not engage in strength exercises.

## 4. Discussion

The results of this study showed that low protein intake was associated with a higher risk of weak HGS in Korean men aged over 60 years relative to adequate protein intake, irrespective of sociodemographic factors, lifestyle factors, and comorbidity. Korean women over the age of 60 who engaged in strength exercises with low protein intake were at a higher risk of having weak HGS compared to those with adequate protein intake. Moreover, the risk of weak HGS was reduced in both Korean men and women over 60 who performed strength exercises and increased their protein intake through their diet.

Maintaining muscle mass and function requires sufficient intake of dietary protein, because the amino acids therein serve as regulators of muscle protein synthesis [9]. Previous research on the relationship between protein intake and HGS in older adults has shown mixed results [18,19,20,21,22]. Some studies found that higher protein intake was associated with a reduction in the loss of muscle strength and function, even in the absence of any difference in initial HGS, based on protein intake levels [18,20]. In another study of 554 older women (65.7–71.6 years), higher protein intake was associated with both higher initial HGS and slower decline in HGS over time [19]. In the present study, men with low protein intake were, relative to those with adequate protein intake, more likely to have a significantly higher OR (1.46; 95% CI: 1.16–1.83) of weak HGS after adjusting for all covariates. Contrastingly, a study of middle-aged and older Korean adults found no relationship between protein intake and low HGS [11]. Factors, such as the age and race of the study population, sample size, confounders, and the measurement and cutoff value for protein intake may explain the differences in the study results.

This study found that in women, unlike in men, low protein intake relative to adequate protein intake was not significantly associated with a higher OR (1.18; 95% CI: 0.93–1.50) of weak HGS, even after adjusting for all covariates. Whereas it is unclear why the results differed between men and women, there are several possible explanations for this gender difference. First, besides the amount of protein, other factors, such as the way protein is distributed, its quality, and its source also can be important for muscle function, though none of them were considered in this study [9]. In this study, around half of the women had low protein intake, while about one-third of the men did. In addition, it has been reported that elderly Korean women consume less than 14 g of protein per meal (compared to 20 g for men) [23], which is lower than the recommended intake of 30 to 40 g per meal for stimulation of muscle protein synthesis [24,25]. Compared with elderly Korean men, elderly Korean women have shown a lower intake of animal protein, which has a more positive effect on muscle growth due to its higher digestibility and higher content of essential amino acids (especially leucine) [23,26]. The second possible explanation for the gender difference in the effects of protein intake is that the protein cutoff used to define adequacy in women is different from that used in men, due to differences in body composition. For both elderly men and women, the recommended dietary allowance for protein is based on nitrogen balance studies [27]. Although the recommended intake is set at 0.8 g per kilogram of body weight per day, research using a stable-isotope-based minimally invasive indicator amino acid oxidation (IAAO) method has found that the optimal intake for older women and men is 1.29 and 1.24 g per kilogram of body weight per day, respectively [28]. Third, the overall quality of diet may be more important than individual nutrient intake, such as that of protein, which possibility was not considered in the current study. In the Newcastle 85+ cohort, a diet high in healthy foods, such as fruits, fish, eggs, nuts, and whole grains was associated with stronger HGS when compared with a diet high in meat and butter [29].

Dietary protein intake combined with resistance exercises has been known to stimulate muscle protein synthesis and to inhibit muscle protein breakdown, leading to the maintenance or even accretion of lean body mass and strength [7]. In this study, the likelihood of weak HGS was significantly reduced for both men and women who engaged in strength training and increased their dietary protein intake (*P_trend_* < 0.001 for men and *P_trend_* = 0.025 for women, respectively) after adjusting for all covariates. In addition, women with low protein intake who engaged in strength exercises had significantly higher odds of weak HGS, that is, 2.46 (95% CI: 1.0.3–5.92)s as compared with women with adequate protein intake after adjusting for all covariates. By contrast, dietary protein intake level without strength exercises was not significantly associated with risk of weak HGS in both men and women. This finding is consistent with a previous randomized controlled trial (RCT) of 280 healthy older adults (aged > 65 years), which showed that recommending protein supplementation as a stand-alone intervention for healthy older individuals was not effective in improving muscle mass and strength [30]. Meta-analyses analyzing RCTs conducted with elderly subjects also have shown that protein supplementation alone is ineffective and that protein supplementation combined with resistance training is more effective in enhancing muscle mass and strength [10,22,31,32]. In the combined exercise and high protein whole food-based diet intervention, dietary intervention enhanced the training-induced improvements in lower limb strength of older adults, compared to an exercise or diet only intervention [33], which also supports the findings of this study.

Protein intake can have anabolic effects, especially after exercise, as muscle protein synthesis in response to amino acids is increased for more than 24 h after resistance exercises [7]. This prolonged response to protein can lead to increased muscle mass and strength [34]. In fact, this may explain why high protein intake with strength exercises seems to have a significant effect on muscle mass and strength.

This study is the first to have investigated the association of dietary protein intake with weak HGS for representative Koreans aged over 60 years in consideration of strength exercise. However, there are some limitations. First, no causal link between protein intake and weak HGS could be established, given the study’s cross-sectional design. Second, due to a lack of information, neither dietary pattern nor the distribution, quality or source of dietary protein possibly affecting HGS was considered. Third, for the same reason, neither duration, frequency, intensity, or specific type of exercise was considered either. Fourth, dietary intake data were collected by the 24-h diet recall method. Accurately assessing the protein intake of study subjects over just a single day is problematic. However, even though self-reported nutritional surveys may be prone to bias or incomplete reporting, which can affect data accuracy, a survey method conducted by a trained interviewer using standardized protocols, as in the present study, could minimize this bias. Fifth and finally, it may be difficult to generalize this study’s results to people of other ethnicities, as it was conducted with an exclusively Korean population.

In conclusion, in this study with a representative sample of Koreans aged over 60, men with low protein intake had a higher risk of weak HGS, as compared with men with adequate protein intake. Meanwhile, men and women with a higher protein intake who also engaged in strength exercises had a lower risk of weak HGS. Increasing protein intake and engaging in strength exercises may be an effective way to preserve muscle strength in older men and women. However, further research in the form of longitudinal or intervention studies would be necessary for confirmation.

## Figures and Tables

**Figure 1 nutrients-15-01014-f001:**
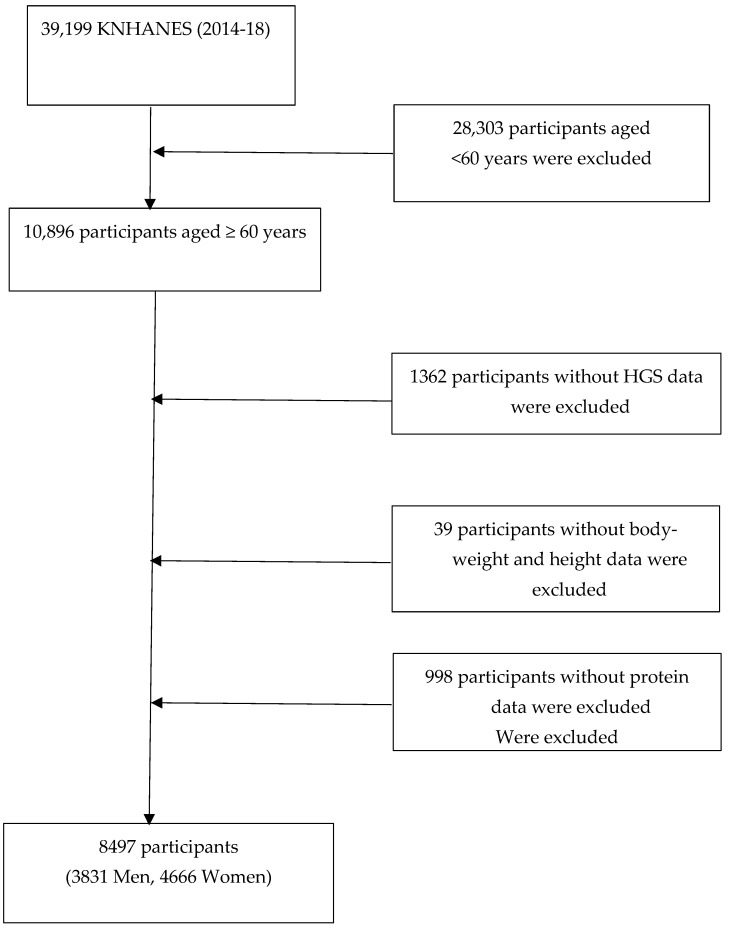
Flow diagram of study participant selection. KNHANES, Korean National Health and Nutrition Examination Survey; HGS, handgrip strength.

**Table 1 nutrients-15-01014-t001:** Demographic characteristics of the participants according to protein intake levels.

		Men				Women		
	Protein Per kg BW			*p*	Protein Per kg BW			*p*
	<0.8	≥0.8 and <1.2	≥1.2		<0.8	≥0.8 and <1.2	≥1.2	
Unweighted sample size	1365	1364	1102		2306	1441	919	
Proportion (%)	34.5 (1.0)	36.5 (0.9)	29.0 (0.9)	<0.001	49.2 (0.9)	30.8 (0.8)	20.0 (0.7)	<0.001
Age, y	70.2 (0.2)	68.9 (0.2)	67.7 (0.2)	<0.001	71.3 (0.2)	69.6 (0.2)	67.5 (0.2)	<0.001
60–69	48.9 (1.6)	57.0 (1.5)	65.8 (1.7)	<0.001	41.7 (1.2)	53.0 (1.6)	66.4 (1.8)	<0.001
70–79	37.2 (1.5)	33.5 (1.4)	28.4 (1.5)		40.4 (1.2)	35.2 (1.5)	26.2 (1.6)	
≥80	13.9 (1.0)	9.5 (0.9)	5.8 (0.8)		17.9 (1.0)	11.8 (1.0)	7.4 (1.0)	
BMI, kg/m^2^	24.5 (0.1)	23.8 (0.1)	23.3 (0.1)	<0.001	25.1 (0.1)	24.1 (0.1)	23.3 (0.1)	<0.001
HGS (kg)	33.3 (0.2)	35.0 (0.2)	35.7 (0.2)	<0.001	20.3 (0.2)	20.8 (0.2)	21.6 (0.2)	<0.001
Protein intake (g/day)	39.3 (0.3)	65.5 (0.4)	104.6 (1.1)	<0.001	32.5 (0.2)	54.8 (0.3)	87.0 (0.9)	<0.001
Protein intake/kg (g/kg)	0.6 (0.00)	1.0 (0.00)	1.6 (0.02)	<0.001	0.6 (0.00)	1.0 (0.00)	1.6 (0.02)	<0.001
Currently smoking				0.091				0.561
no	75.3 (1.5)	79.3 (1.3)	78.4 (1.4)		97.4 (0.4)	97.6 (0.5)	98.2 (0.6)	
yes	24.7 (1.5)	20.7 (1.3)	21.6 (1.4)		2.6 (0.4)	2.4 (0.5)	1.8 (0.6)	
Heavy drinking				0.014				0.001
no	89.0 (1.1)	89.2 (1.1)	84.8 (1.3)		99.3 (0.2)	98.8 (0.3)	97.6 (0.6)	
yes	11.0 (1.1)	10.8 (1.1)	15.2 (1.3)		1.7 (0.2)	1.2 (0.3)	2.4 (0.6)	
Education				<0.001				<0.001
<university	77.2 (1.4)	66.8 (1.6)	64.2 (1.7)		96.8 (0.48)	92.2 (0.8)	86.5 (1.3)	
≥university	22.8 (1.4)	33.2 (1.6)	35.8 (1.7)		3.2 (0.4)	7.8 (0.8)	13.5 (1.3)	
Aerobic PA				<0.001				<0.001
no	64.6 (1.6)	57.8 (1.7)	53.3 (1.7)		71.6 (1.2)	65.8 (1.5)	62.0 (1.9)	
yes	35.4 (1.6)	42.2 (1.7)	46.7 (1.7)		28.4 (1.2)	34.2 (1.5)	38.0 (1.9)	
Strength exercise			<0.001				<0.001
no	77.2 (1.4)	66.8 (1.6)	64.2 (1.7)		91.6 (0.7)	86.4 (1.1)	82.8 (1.5)	
yes	22.8 (1.4)	33.2 (1.6)	35.8 (1.7)		8.4 (0.7)	13.6 (1.1)	17.2 (1.5)	
Chronic disease				<0.001				<0.001
no	61.3 (1.7)	63.3 (1.6)	70.9 (1.4)		61.5 (1.3)	68.7 (1.5)	72.4 (1.8)	
yes	38.7 (1.7)	36.7 (1.6)	29.1 (1.4)		38.5 (1.3)	31.3 (1.5)	27.6 (1.8)	
Limitation				<0.001				<0.001
no	83.6 (1.2)	88.5 (1.0)	91.7 (0.9)		79.7 (1.1)	85.5 (1.1)	89.0 (1.3)	
yes	16.4 (1.2)	11.5 (0.9)	8.3 (0.9)		20.3 (1.1)	14.5 (1.1)	11.0 (1.3)	
Occupation				<0.001				<0.001
White collar worker	37.8 (1.6)	48.0 (1.9)	49.4 (1.9)		40.2 (1.4)	48.5 (1.7)	51.6 (2.2)	
Blue collar worker	62.2 (1.6)	52.0 (1.9)	50.6 (1.9)		59.8 (1.4)	51.5 (1.7)	48.4 (2.2)	
Weak HGS				<0.001				<0.001
no	76.3 (1.3)	83.3 (1.1)	85.9 (1.2)		66.4 (1.4)	73.6 (1.4)	79.5 (1.5)	
yes	23.7 (1.3)	16.7 (1.1)	14.1 (1.2)		33.6 (1.4)	26.4 (1.4)	20.5 (15)	

Values are represented as weighted means or weighted proportions with standard error for continuous or categorical variables. *p* was determined by complex sample cross-tab analysis or linear regression analysis. Abbreviations; BMI, body mass index; BW, body weight; HGS, handgrip strength; PA, physical activity.

**Table 2 nutrients-15-01014-t002:** Association of weak HGS with protein intake level according to gender.

	Protein Intake Per kg BW				
	<0.8	≥0.8 and <1.2	≥1.2	*p*	*P* * _trend_ *
Men					
Prevalence (%)	23.7 (1.3)	16.7 (1.1)	14.1 (1.2)	<0.001	<0.001
Model I	1.46 (1.16–1.84)	1.00	0.93 (0.71–1.22)	0.001	0.001
Model II	1.46 (1.16–1.83)	1.00	1.01 (0.73–1.38)	0.010	0.035
Women					
Prevalence (%)	33.6 (1.4)	26.4 (1.4)	20.5 (1.1)	<0.001	<0.001
Model I	1.18 (0.97–1.44)	1.00	0.94 (0.74–1.20)	0.093	0.032
Model II	1.18 (0.93–1.50)	1.00	0.96 (0.74–1.27)	0.270	0.118

Values are expressed as percent (SE) or ORs (with 95% CIs) as derived by the complex sample cross-tab analysis or complex sample logistic regression modeling. Model I: adjusted for age and body mass index; Model II: adjusted for all variables in model I plus education, aerobic physical activity, strength exercise, chronic disease, limitation, heavy alcohol use, occupation, and smoking status. Abbreviations: BW, body weight; HGS, handgrip strength; SE, standard error; ORs, odds ratios; CIs, confidence intervals.

**Table 3 nutrients-15-01014-t003:** Associations of weak HGS with protein intake level according to gender and strength exercises.

	Protein Intake Per kg BW				
	< 0.8	≥0.8 and <1.2	≥1.2	*p*	*P* * _trend_ *
Men					
Strength exercises					
No	1.36 (1.01–1.82)	1.00	1.14 (0.80–1.62)	0.121	0.252
Yes	1.77 (0.88–3.54)	1.00	0.67 (0.34–1.32)	0.008	<0.001
Women					
Strength exercises					
No	1.11 (0.87–1.43)	1.00	0.96 (0.72–1.28)	0561	0.288
Yes	2.46 (1.03–5.92)	1.00	0.91 (0.35–2.37)	0.050	0.025

Values are expressed as ORs (with 95% CIs) derived by the complex sample logistic regression modeling adjusted for age, BMI, education, aerobic physical activity, chronic disease, limitation, heavy alcohol use, occupation, and smoking status. Abbreviations: BW, body weight; HGS, handgrip strength; ORs, odds ratios; CIs, confidence intervals.

## Data Availability

Data are available at https://knhanes.cdc.go.kr/ (accessed on 1 November 2022).

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
