# Peer review of "Association of Protein Intake with Handgrip Strength and Its Relation to Strength Exercise in Korean Adults Aged over 60 Years in the KNHANES (2014-18)"

_nutrients, 2023, doi:10.3390/nu15041014_

Round 1
Reviewer 1 Report
The authors of the paper undertook the topic of evaluating the relationship between protein intake and muscle strength. The topic is part of the wide interest in sarcopenia risk factors and actions aimed at counteracting this deficit. Although the topic has been frequently discussed in recent years, the value of the work lies in the population nature of the data, concerning a large group of elderly people in a homogeneous Korean population.
My comments on the manuscript:
1. Abstract: The presented data concern both sexes, therefore, despite the fact that the assessed relationship was not statistically significant in women, the abstract should include information on this result.
2. Introduction: I agree that the general recommendations recommend 0.8g/kg of protein for adults, but since the nutritional guidelines have been recommending 1.0-1.2g/kg of protein for the elderly for years, I suggest citing a literature source for these general recommendations.
3. Methods section:
- description of physical activity assessment is not clear – it suggests the use of IPAQ-short form (?). The phrase "Aerobic exercise was defined as engaging in vigorous activity for at least 20 minutes at least three days per week, or moderate exercise/walking for at least 30 minutes at least five days per week, based on the International Physical Activity Questionnaire." should be reformulated - rather it is not about defining aerobic effort, but about its gradation.
- How was involvement in strength training assessed?
- I admit that I do not know the prevalence of chronic diseases in the older Korean population, but I am surprised that chronic diseases such as COPD, heart failure, and thyroid diseases are not included. Were these diseases sporadic?
- In the description of the statistical analysis, when describing the division of the subjects in terms of protein intake, it was written: “The participants were divided into three 107 groups based on protein intake (<0.8, 0.8-1.2, ≥1.2 g/kg),” that is, the value of 1.2 was in two subgroups - needs correction.
4. Result section:
- The main remark concerns the error in the presentation of the tables: no correct table 2, the table presented as 2 is probably table 3, and the table 3 should probably be table 4. As a result, the manuscript contains two tables with the number 3 and table 2 and one of the tables with number 3 are identical.
- Explanations of the regression models (models 1 and 2) are in the wrong place, close to table 2.
- Since the average age of individual analyzed groups is not high, and muscle strength is quite well preserved, in order to present the age structure of the studied population, it is worth specifying the age range of the respondents and the percentage of the oldest people (85+) in individual groups.
5. Discussion:
- Comparing the percentages of men and women involved in strength training shows that significantly fewer women exercise - is it possible that this factor, according to the arguments in the discussion, related to the lack of a significant relationship between protein intake and muscle strength in women, especially since the average values of protein per kg body weights are similar in groups of women and men.
6. Conclusion: As the statistically significant result applies only to men, the sentence “Increasing protein intake and 275 engaging in strengthening exercise may be an effective way to preserve muscle strength 276 in older adults” is too much of a generalization and needs to be reformulated.
Author Response
- Abstract: The presented data concern both sexes, therefore, despite the fact that the assessed relationship was not statistically significant in women, the abstract should include information on this result.
->I appreciate your valid point. As per your suggestion, I added the sentence “this relationship was not statistically significant in women.” in the result of abstract.
- Introduction: I agree that the general recommendations recommend 0.8g/kg of protein for adults, but since the nutritional guidelines have been recommending 1.0-1.2g/kg of protein for the elderly for years, I suggest citing a literature source for these general recommendations.
--> As per your suggestion, I added a reference (No.7) for these general recommendations.
- Methods section:
- description of physical activity assessment is not clear – it suggests the use of IPAQ-short form (?). The phrase "Aerobic exercise was defined as engaging in vigorous activity for at least 20 minutes at least three days per week, or moderate exercise/walking for at least 30 minutes at least five days per week, based on the International Physical Activity Questionnaire." should be reformulated - rather it is not about defining aerobic effort, but about its gradation.
--> I appreciate your valid point. The Korean National Health and Nutrition Examination Survey (KNHANES) used the shortened version of the International Physical Activity Questionnaire (IPAQ) until 2013, and has utilized Global Physical Activity Questionnaire since 2014. It was entirely my responsibility that I failed to notice that the Global Physical Activity Questionnaire had been used since 2014. I am grateful to the reviewer for identifying this significant error. "The term aerobic exercise was altered to aerobic physical activity, and its definition was further revised as follows:
Aerobic physical activity was defined as engaging in 150 minutes of moderate-intensity physical activity or 75 minutes of vigorous-intensity physical activity or an equivalent combination of moderate- and vigorous-intensity physical activity based on the Global Physical Activity Questionnaire.
- How was involvement in strength training assessed?
--> The participant who reported engaging in strength exercises, such as push-ups, sit-ups, using dumbbells or weights, or using a horizontal bar more than twice in the past week, was considered to be participating in strength exercise. It was described in the methods.
- I admit that I do not know the prevalence of chronic diseases in the older Korean population, but I am surprised that chronic diseases such as COPD, heart failure, and thyroid diseases are not included. Were these diseases sporadic?
--> I appreciate your valid point. I apologize that COPD, heart failure and thyroid disease were not included in the chronic disease. Unfortunately, the lack of a questionnaire on diagnosis of heart failure resulted in the exclusion of heart failure from the category of chronic diseases. As asthma is also one of the chronic diseases, asthma, COPD and thyroid diseases were included in the chronic disease. New analysis were presented in the text and tables.
- In the description of the statistical analysis, when describing the division of the subjects in terms of protein intake, it was written: “The participants were divided into three 107 groups based on protein intake (<0.8, 0.8-1.2, ≥1.2 g/kg),” that is, the value of 1.2 was in two subgroups - needs correction.
--> I am grateful for bringing the error to my attention. It has been revised as “≥0.8 and <1.2” and marked in the text and table.
- Result section:
- The main remark concerns the error in the presentation of the tables: no correct table 2, the table presented as 2 is probably table 3, and the table 3 should probably be table 4. As a result, the manuscript contains two tables with the number 3 and table 2 and one of the tables with number 3 are identical.
--> I apologize that the tables are not presented correctly. As you anticipated, the table presented as 2 is table 3, and table presented as 3 corresponds to table 2. The table that was placed incorrectly has been corrected
- Explanations of the regression models (models 1 and 2) are in the wrong place, close to table 2.
--> I apologize that the explanations of regression models are not presented correctly. It was modified and now has been placed under table 2.
- Since the average age of individual analyzed groups is not high, and muscle strength is quite well preserved, in order to present the age structure of the studied population, it is worth specifying the age range of the respondents and the percentage of the oldest people (85+) in individual groups.
--> I apologize that I could not give you the exact percentage of the oldest group (85+) in each category because the National Health and Nutrition Survey only records individuals over 80 as being 80 years old. Instead, age group percentage was presented in table 1 as follows.
|
  |
  |
Male |
  |
  |
  |
  |
Female |
  |
  |
|
 Protein per kg BW  |
P |
  Protein per kg BW  |
P |
||||||
|
  |
<0.8 |
0.8≤ <1.2 |
≥1.2 |
  |
  |
<0.8 |
0.8≤ <1.2 |
≥1.2 |
  |
|
60-69 |
48.9 (1.6) |
57.0 (1.5) |
65.8 (1.7) |
<0.001 |
41.7 (1.2) |
53.0 (1.6) |
66.4 (1.8) |
<0.001 |
|
|
70-79 |
37.2 (1.5) |
33.5 (1.4) |
28.4 (1.5) |
40.4 (1.2) |
35.2 (1.5) |
26.2 (1.6) |
|||
|
≥80 |
13.9 (1.0) |
9.5 (0.9) |
5.8 (0.8) |
  |
  |
17.9 (1.0) |
11.8 (1.0) |
7.4 (1.0) |
  |
- Discussion:
- Comparing the percentages of men and women involved in strength training shows that significantly fewer women exercise - is it possible that this factor, according to the arguments in the discussion, related to the lack of a significant relationship between protein intake and muscle strength in women, especially since the average values of protein per kg body weights are similar in groups of women and men.
--> The updated analysis revealed a significant relationship between protein intake level and weak HGS in women who engaged in strength exercise. It was not believed that the reduced participation of women in strength exercise would have a negative effect on the results.
- Conclusion: As the statistically significant result applies only to men, the sentence “Increasing protein intake and 275 engaging in strengthening exercise may be an effective way to preserve muscle strength 276 in older adults” is too much of a generalization and needs to be reformulated.
--> I appreciate your valuable suggestion. Based on the updated analysis that showed a significant decrease in the risk of weak HGS for both men and women who engaged in strength exercise and had high protein intake, the conclusion remained unchanged.
Reviewer 2 Report
The authors investigated into the relationship between low handgrip and protein intake in a sample of the KNHANES Study.
Overall, there is already a paper published with the topic of low handgrip in 2019 by Kim et al. also including the KNHANES (2014-2016) population and nutrition as well.the reviewer is aware that the sample now in the submitted manuscript includes a longer time period.
Nevertheless, several major concerns needed to be addressed before the manuscript can be rated for publication.
Major concerns:
Title: As the author is also taking strength exercise into the whole manuscript as an important variable it should be mentioned in the title as well.
Key words:
for the reviewer the key words of Korean AND KNHANES seem to overlap.
Introduction:
The introduction is very short and does not include all necessary information for the later analyses. References for the interaction of low handgrip and exercise e.g. need some information on what type of strength exercise ( resistance or power or vibration training), the length of the exercise period having an effect on this interaction, and being performed with progression. The overall infomration on the relationship between low protein and and low handgrip is not necessarily so new therefore, taking strength exercise into this relationship is the new aspect of this manuscript and needs strengthening.
Methods:
the reviewer wants to congratulate the author for the impressive sample. The KNHANES study is of high quality and the included number of participants are remarkable. the reviewer only wonders if participants with hand arthritis were included in the analysis or if they belonged to the group of wrist pain. Please explain.
AS HG is presented only in max HG (being the capacity measure) the reviewer wonders if the functional HG (mean of all three trials) would give other results but could be of clinical higher relevance.
Another question the reviewer has: is there an impact of being a "white" or "blue" colour worker during life? Does the KNHANES data pool provide any information on this? The reviewer -out of experience form own research - knows that e.g. farmers have high HG due to their daily life.
Results:
One of the major concerns the reviewer have is the length of taking part in strengthening exercise. Information on this information is crucial this aspect has an impact of the association.
Discussion:
The reviewer wants to congratulate the author for the discussion on the possible explanation on gender differences of the results.
Whereas, the discussion point of results of strength training in combination with protein supplementation is not related to the topic of this manuscript including no protein supplementation. This needs to be corrected as the cited references are of intervention but not normal nutrition intake.
Please also add a short paragraph about the strength and limitations of the study.
Minor Concerns:
the reviewer is not sure if the comment is applicable but in the abstract there seems to be different format of writing. In addition, please review the manuscript for minor spelling mistakes.
In addition the formatting regarding around the Tables seemed also to be impaired.
Author Response
Title: As the author is also taking strength exercise into the whole manuscript as an important variable it should be mentioned in the title as well.
response: I appreciate your suggestion. So, the title is changed as “the association of dietary protein intake with HGS and its relation to strength exercise in Korean adults aged more than 60 years in KNHANES (2014-18)
Key words:
for the reviewer the key words of Korean AND KNHANES seem to overlap.
response: KNHANES was deleted as per your opinion.
Introduction:
1.The introduction is very short and does not include all necessary information for the later analyses. References for the interaction of low handgrip and exercise e.g. need some information on what type of strength exercise ( resistance or power or vibration training), the length of the exercise period having an effect on this interaction, and being performed with progression. The overall information on the relationship between low protein and low handgrip is not necessarily so new therefore, taking strength exercise into this relationship is the new aspect of this manuscript and needs strengthening.
response: I appreciate your valuable opinion. According to your opinion, I modified the introduction focusing on effect of both strength exercise and protein intake on HGS.
Methods:
- the reviewer wants to congratulate the author for the impressive sample. The KNHANES study is of high quality and the included number of participants are remarkable. the reviewer only wonders if participants with hand arthritis were included in the analysis or if they belonged to the group of wrist pain. Please explain.
response: Thank you for your good question. It's challenging to provide a precise answer about your question because the raw data of inspection and survey of grip strength tests, including information on participants' hand arthritis diagnosis, is not accessible to the public. It is thought that patients with hand arthritis with wrist pain were excluded, but it is not clear whether patients with hand arthritis without wrist pain were excluded.
- AS HG is presented only in max HG (being the capacity measure) the reviewer wonders if the functional HG (mean of all three trials) would give other results but could be of clinical higher relevance.
response: Thank you for your good suggestion. I tried to analyze the data as per your recommendation of functional HGS (mean of all three trials). Although prevalence of weak HGS increased, weak HGS according to functional HGS does not give any significant results.
<Association of weak HGS according to functional HG with protein intake level according to gender.>
|
  |
Protein intake per kg BW |
  |
  |
||
|
  |
<0.8 |
≥0.8 and <1.2 |
≥1.2 |
P |
Ptrend |
|
Men |
  |
  |
|||
|
Prevalence (%) |
30.7 (1.4) |
23.6 (1.3) |
19.8 (1.3) |
<0.001 |
<0.001 |
|
Model I |
1.35 (1.09-1.67) |
1.00 |
0.89 (0.70-1.12) |
0.001 |
0.001 |
|
Model II |
1.24 (0.98-1.58) |
1.00 |
0.92 (0.70-1.21) |
0.072 |
0.035 |
|
Women |
|||||
|
Prevalence (%) |
43.4 (1.4) |
36.8 (1.5) |
30.6 (1.7) |
<0.001 |
<0.001 |
|
Model I |
1.12 (0.93-1.34) |
1.00 |
0.96 (0.77-1.19) |
0.285 |
0.120 |
|
Model II |
1.15 (0.93-1.42) |
1.00 |
0.99 (0.78-1.26) |
0.356 |
0.118 |
Model I: adjusted for age and body mass index; Model II: adjusted for all variables in model I plus education, aerobic physical activity, strength exercise, chronic disease, limitation, heavy alcohol use, occupation, and smoking status.
Abbreviations: BW, body weight; HGS, handgrip strength; SE, standard error; ORs, odds ratios; Cis, confidence intervals.
- Another question the reviewer has: is there an impact of being a "white" or "blue" colour worker during life? Does the KNHANES data pool provide any information on this? The reviewer -out of experience form own research - knows that e.g. farmers have high HG due to their daily life.
response: I appreciate your curiosity and ideas The KNHANES data contains information about occupation. Occupation was divided into two categories: white-collar workers and blue-collar workers. White-collar workers included managers, professionals, clerks, and sales workers. Blue-collar workers included skilled workers in agriculture, forestry, fishing, craft trades, machine operation, and elementary labor. In multiple logistic regression, blue color workers had significantly higher odds ratio of weak HGS after adjusting all covariates: OR 1.42 (95% CI 1.09-1.85 p=0.010) for men and OR 1.25 (95% CI 1.02-1.53 p=0.032) for women. As occupation was significantly associated with weak HGS, occupation was also included in the multiple logistic model. The proportion of occupation according to dietary protein intake level is displayed in table 1 and the results are as follows.
|
  |
  |
Men |
  |
  |
  |
  |
Women |
  |
  |
|
 Protein per kg BW  |
P |
 Protein per kg BW  |
P |
||||||
|
  |
<0.8 |
≥0.8 and <1.2 |
≥1.2 |
  |
  |
<0.8 |
≥0.8 and <1.2 |
≥1.2 |
  |
|
Occupation |
|
|
|
<0.001 |
|
|
|
|
<0.001 |
|
White collar worker |
37.8 (1.6) |
48.0 (1.9) |
49.4 (1.9) |
|
|
40.2 (1.4) |
48.5 (1.7) |
51.6 (2.2) |
|
|
Blue collar worker |
62.2 (1.6) |
52.0 (1.9) |
50.6 (1.9) |
|
|
59.8 (1.4) |
51.5 (1.7) |
48.4 (2.2) |
|
One of the major concerns the reviewer have is the length of taking part in strengthening exercise. Information on this information is crucial this aspect has an impact of the association.
response: I apologized that the length of taking part in strengthening exercise was not included in the survey, making it challenging to determine. In this study, the participant who reported engaging in strength exercises, such as push-ups, sit-ups, using dumbbells or weights, or using a horizontal bar more than twice in the past week, was considered to be participating in strength exercise. I brought attention to this concern in the limitations part of the paper.
Discussion:
- The reviewer wants to congratulate the author for the discussion on the possible explanation on gender differences of the results. Whereas, the discussion point of results of strength training in combination with protein supplementation is not related to the topic of this manuscript including no protein supplementation.
response: I’m grateful for your good point. Even though this study wasn't a study that involved the use of protein supplements as an intervention, as I believed the findings of intervention study would reinforce the impact of strength exercise on both groups - those who consumed high amounts of protein with supplements and those who did not. I incorporated additional study that utilized a typical diet as the protein intervention into the discussion.
- Please also add a short paragraph about the strength and limitations of the study.
response: The strength and limitations of the study were already discussed in the preceding paragraph of the conclusion.
Minor Concerns:
- the reviewer is not sure if the comment is applicable but in the abstract there seems to be different format of writing.
response: I apologize for the oversight in missing the change in writing format and for any inconvenience it may have caused. The issue has been corrected.
- In addition, please review the manuscript for minor spelling mistakes.
response: I appreciate your meticulous attention. I reviewed and made modifications.
- In addition, the formatting regarding around the Tables seemed also to be impaired.
response: I am grateful for your meticulous attention. The formatting around the Tables was corrected.
Round 2
Reviewer 1 Report
After correction of manuscript I am satisfied, but in the Discussion, the Author has not changed the odds ratio data for men and women according to the new analysis: on page 8 in line 227 and in line 233 the data from the previous version are still there.
In response to my comments, the author posted interesting data on the percentage of people with low protein intake depending on age. They clearly show that protein intake decreases with age - it's a pity that this was not included in the article. I understand that these data do not fit into the analyzed topic.